# Optimizing Neural Networks with Gradient Lexicase Selection

**Li Ding**
University of Massachusetts Amherst
liding@umass.edu

**Lee Spector**
Amherst College
University of Massachusetts Amherst
lspector@amherst.edu

## Abstract

One potential drawback of using aggregated performance measurement in machine learning is that models may learn to accept higher errors on some training cases as compromises for lower errors on others, with the lower errors actually being instances of overfitting. This can lead both to stagnation at local optima and to poor generalization. Lexicase selection is an uncompromising method developed in evolutionary computation, which selects models on the basis of sequences of individual training case errors instead of using aggregated metrics such as loss and accuracy. In this paper, we investigate how the general idea of lexicase selection can fit into the context of deep learning to improve generalization. We propose Gradient Lexicase Selection, an optimization framework that combines gradient descent and lexicase selection in an evolutionary fashion. Experimental results show that the proposed method improves the generalization performance of various popular deep neural network architectures on three image classification benchmarks. Qualitative analysis also indicates that our method helps the networks learn more diverse representations.

## 1 Introduction

Modern data-driven learning algorithms, in general, define an optimization objective, *e.g.*, a fitness function for parent selection in genetic algorithms (Holland, 1992) or a loss function for gradient descent in deep learning (LeCun et al., 2015), which computes the aggregate performance on the training data to guide the optimization process. Taking the image classification problem as an example, most recent approaches use Cross-Entropy loss with gradient descent (Bottou, 2010) and backpropagation (Rumelhart et al., 1985) to train deep neural networks (DNNs) on batches of training images. Despite the success that advanced DNNs can reach human-level performance on the image recognition task (Russakovsky et al., 2015), one potential drawback for such aggregated performance measurement is that the model may learn to seek "compromises" during the learning procedure, *e.g.*, optimizing model weights to intentionally keep some errors in order to gain higher likelihood on correct predictions. To give an example, consider a situation that may happen during the training phase of image classification for a batch of 10 images: 9 of them are correctly predicted with high probabilities, but one is wrong. The aggregated loss may produce gradients that guide the model weights to compromise the wrong case for higher probabilities on other cases, which may lead to the optimization process getting stuck at local optima.

We refer to problems for which such compromises are undesirable as *uncompromising problems* (Helmuth et al., 2014), that is, as problems for which it is not acceptable for a solution to perform sub-optimally on any one of the cases in exchange for better performance on others. In deep learning, in order to improve the generalization (Zhang et al., 2017) of DNNs, it is important to maintain the diversity and generality of the representations contributed by every training case.

From the literature, uncompromising problems have been recently explored in genetic programming (GP) and genetic algorithms (GAs) for tasks such as program synthesis. Among many methods that aim to mitigate this problem, lexicase selection (Helmuth et al., 2014; Spector, 2012) has been shown to outperform many other methods (Fieldsend & Moraglio, 2015; Galvan-Lopez et al., 2013; Krawiec & Liskowski, 2015) in a number of applications and benchmarks (Helmuth & Spector,

2015; Helmuth & Kelly, 2021). Instead of using an aggregated fitness function for parent selection, lexicase selection gradually eliminates candidates as it proceeds to look at how the population fares at each data point in the shuffled training dataset, in which way it can bolster the diversity and generality in populations. Recent works also show that lexicase selection can be used in rule-based learning systems (Aenugu & Spector, 2019), symbolic regression (La Cava et al., 2016), constraint satisfaction problems (Metevier et al., 2019), machine learning (La Cava & Moore, 2020b;a), and evolutionary robotics (Huizinga & Clune, 2018; La Cava & Moore, 2018) to improve model generalization, especially in situations of diverse and unbalanced data. It is reasonable to suspect that for many deep learning problems such as image classification, due to natural variances in real-world data collection, lexicase selection is likely to help improve the generalization of models.

In this work, we aim to explore the application of lexicase selection in the task of optimizing deep neural networks. Taking advantage of the commonly-used gradient descent and backpropagation methods, we introduce Gradient Lexicase Selection, an optimization framework for training deep neural networks that not only benefit from the efficiency of gradient-based learning but also improves the generalization of the networks using the outline of lexicase selection method in an evolutionary fashion. We test the proposed method on the basic image classification task on three benchmark datasets (CIFAR-10 (Krizhevsky et al., 2009), CIFAR-100 (Krizhevsky et al., 2009), and SVHN (Netzer et al., 2011)). Experimental results show that gradient lexicase selection manages to improve the performance of the DNNs consistently across six different popular architectures (VGG (Simonyan & Zisserman, 2015), ResNet (He et al., 2016), DenseNet (Huang et al., 2017), MobileNetV2 (Sandler et al., 2018), SENet (Hu et al., 2018), EfficientNet (Tan & Le, 2019)). In addition, we perform further ablation studies to analyze the effectiveness and robustness of the proposed method from various perspectives. We first introduce variants to our method by using random selection and tournament selection, in order to validate the contribution of each component in the framework. We also investigate the trade-offs between exploration and exploitation by analyzing the effects of changing population size and momentum. Finally, the qualitative analysis shows that our algorithm can produce better representation diversity, which is advantageous to the generalization of DNNs.

## 2 BACKGROUND AND RELATED WORK

**Preliminaries of Lexicase Selection**

Lexicase selection is initially proposed as a parent selection method in population-based stochastic search algorithms such as genetic programming (Helmuth et al., 2014; Spector, 2012). Follow-up work has shown that lexicase selection can effectively improve behavioral diversity and the overall performance and on a variety of genetic programming problems (Helmuth et al., 2016; Helmuth & Spector, 2015; Helmuth et al., 2014; Liskowski et al., 2015). The key idea in lexicase selection is that each selection event considers a randomly shuffled sequence of training cases. As a result, lexicase selection sometimes selects specialist individuals that perform poorly on average but perform better than many individuals on one or more other cases. A more detailed description of lexicase selection is appended in Sec. A.

Unlike methods such as tournament selection that use a single fitness value and thus tend to always select generalist individuals that have good average performance, lexicase selection does not base selection on an aggregated measure of performance. Such a difference allows lexicase selection to maintain higher population diversity by prioritizing different parts of the dataset during each selection event through the ordering of the cases. It has been shown empirically on a number of program synthesis benchmark problems that lexicase selection substantially outperforms standard tournament selection and typically maintains higher levels of diversity (Helmuth et al., 2016).

In a more general context, lexicase selection can be used in any case where a selection procedure occurs with regard to performance assessment of multiple candidates with a set of training cases. Recent work also explores the usage of lexicase selection in rule-based learning systems (Aenugu & Spector, 2019), symbolic regression (La Cava et al., 2016), constraint satisfaction problems (Metevier et al., 2019), machine learning (La Cava & Moore, 2020b;a), and evolutionary robotics (Huizinga & Clune, 2018; La Cava & Moore, 2018). In this work, we aim to explore the effectiveness of lexicase selection in the context of deep learning optimization from the perspective of improving the diversity of gradient-based representation learning for better generalization. While

there are also other parent selection methods (Fieldsend & Moraglio, 2015; Galvan-Lopez et al., 2013; Krawiec & Liskowski, 2015) that have been proposed to achieve similar goals, in this work we focus on investigating the usage of lexicase selection in deep learning. Further discussion of comparisons between lexicase selection to other selection methods are out of the scope of this work.

**Deep Neuroevolution and Population-based Optimization**

While backpropagation (Rumelhart et al., 1985) with gradient descent has been the most successful method in training DNNs with fixed-topology in the past few decades (LeCun et al., 2015), there are also attempts to train DNNs through evolutionary algorithms (EAs). Such et al. (2017) proposed a gradient-free method to evolve the network weights by using a simple genetic algorithm, and was able to evolve a relatively deep network (with 4 million parameters) and demonstrated competitive results on several reinforcement learning benchmark problems. Jaderberg et al. (2017) proposed population-based training to optimize both model and the hyperparameters. Cui et al. (2018) proposed to alternate between the SGD step and evolution step to improve the average fitness of the population. Ding & Spector (2021) demonstrate the usage of selection-inspired methods as regularization of DNNs. The most relevant recent work is Pawełczyk et al. (2018), which did a pilot study on combining simple GA schema with gradient-based learning, where gradient training is used as part of the mutation process. Although the method was tested only on one dataset, the results were encouraging and offered some insights on a deeper combination of GAs and gradient learning. Following this trend, this paper aims to explore a more efficient evolutionary framework that takes advantage of both SGD and lexicase selection, to improve the network generalization by treating image recognition as an uncompromising problem.

From a broader perspective that is also closely related to this work, there has been a surge of interest in methods for Neural Architecture Search (NAS) (Elsken et al., 2019), where evolutionary algorithms gain high popularity. A majority of methods (Floreano et al., 2008; Liu et al., 2017; Miikkulainen et al., 2019; Real et al., 2019; 2017; Stanley & Miikkulainen, 2002; Xie & Yuille, 2017) use EA to search neural network topologies and use backpropagation to optimize network weights, and some others (Stanley & Miikkulainen, 2002) use EA to co-evolve topologies along with weights. While our method can be easily extended to the NAS problem, this work focuses on training various fixed-topology networks in order to make fair comparisons to show the significance of using lexicase selection to improve model generalization.

## 3 GRADIENT LEXICASE SELECTION

Our goal is to integrate lexicase selection to improve the generalization of DNNs, while at the same time to the greatest extent keep the efficiency of the popular gradient-based learning. We propose Gradient Lexicase Selection as an optimization framework to combine the strength of these two methods. The algorithm is outlined in Alg. 1. An overview of the algorithm is also depicted in Fig. 1. The proposed algorithm has two main components, Subset Gradient Descent (SubGD) and Lexicase Selection, which we describe in details in this section as follows.

### 3.1 EVOLUTION WITH SUBSET GRADIENT DESCENT (SUBGD)

First, we introduce the general evolutionary framework that uses a combination of stochastic gradient descent and evolution. Given the network topology, we first initialize all the parameters $\mathbb{W}_0$ as the initial parent weights. For each generation, given a population size of $p$, we generate $p$ instances of the model as $p$ offspring with the same weights as the parent weights, namely, $\mathbb{W}^{(0)} = \mathbb{W}^{(1)} = \cdots = \mathbb{W}^{(p-1)} = \mathbb{W}_0$. We then perform mutation on these offspring and use lexicase selection to select the parent for the next generation.

For each generation, we have $p$ instances of the model starting with the exact same weights. Instead of random mutations such as adding gaussian noise as commonly used in neuroevolution, we propose a gradient-based mutation method called subset gradient descent (SubGD).

Given the whole training dataset $\mathbb{S}_{train}$, we divided it into $p$ subsets with random sampling, as $\mathbb{S}_{train}^{(0)}, \mathbb{S}_{train}^{(1)}, \cdots, \mathbb{S}_{train}^{(p-1)}$. We then train each model instance accordingly on one of the subsets using the normal mini-batch stochastic gradient descent. The mutation is done when all the training data is consumed, which is one epoch in traditional deep learning.

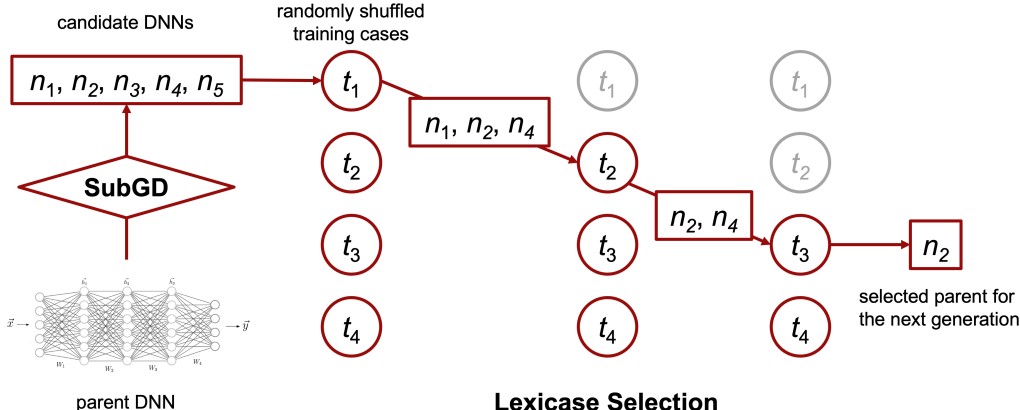

Figure 1: Overview of the proposed gradient lexicase selection. Given the parent model, we first generate candidates by running subset gradient descent (SubGD), then perform lexicase selection by assessing candidates on each individual case to obtain the parent model for the next generation.

There are several advantages of the proposed mutation method. First, since all the offspring are trained with different non-overlapping training samples, they are likely to evolve diversely, especially when data augmentation is also included. Secondly, each off-spring is trained using gradient descent, meaning they will be optimized efficiently towards the objective, comparing to random mutation methods such as gaussian noise. Thirdly, if implemented with distributed training, all the offspring can be trained simultaneously to further reduce computation time. In general, the subset gradient descent aims to find a balance between exploration and exploitation during the evolution process for more efficient optimization.

## 3.2 LEXICASE SELECTION FOR DNNS

After mutation, the offspring become candidates and we use lexicase selection to select a parent from them for the next generation. First, a randomly shuffled sequence of training data points (without data augmentation) is used for selection. Starting from the first training sample, we evaluate all the candidates on each case individually and remove the candidate from the selection pool if it does not make the correct prediction. This process is repeated until if 1) there is only one candidate left, which will be selected as the parent for the next generation, or 2) all the training samples are exhausted and more than one candidates survive, in which case we randomly pick a candidate from the selection pool.

For the selection process, we do not hold out another validation set because 1) if we choose to use a validation set, the validation set should have an adequate size in order to ensure its diversity and generality, which means the training set will be noticeably smaller, and thus the training performance is likely to degrade; 2) since each model instance only gets access to part of the (augmented) training data, the selection performed on the original training data is still effective, since the exact same data was never used in the mutation (training).

An important feature of lexicase selection is that it treats all the cases equally and thus there is no way to modify its selection pressure. The motivation behind lexicase selection is to allow the survival of those models which may not perform best overall but were able to solve the given testing cases. Such a model is likely to learn essential feature representations that allow it to make correct predictions on specific cases where all others fail. By letting lexicase selection guide the training process, the neural network can potentially learn more diverse representations that finally contribute to better generalization.

To better accommodate the situation of training deep models on large-scale datasets, we also make some slight modifications to the original lexicase selection algorithm in regard to the tie situations, *i.e.*, when all the remaining candidates fail to make the correct prediction on one case. The original

---

**Algorithm 1:** Gradient Lexicase Selection

---

**Data:**

- `data` - the whole training dataset
- `candidates` - set of $p$ instances of the DNN model initialized with the same parameters

**Result:**

- an optimized DNN model

```
// K training epochs
```
**for** $epoch = 1 : K$ **do**

   `subsets` $\leftarrow p$ equal-size subsets obtained through random sampling from the entire `data` without replacement

   Use gradient descent and backpropagation to optimize each of the $p$ `candidates` on each of the $p$ `subsets` respectively

   `cases` $\leftarrow$ randomly shuffled sequence of `data` to be used in lexicase selection

   `parent` $\leftarrow$ *None*

   **for** *case in cases* **do**

      Evaluate all the `candidates` on `case`.

      `candidates` $\leftarrow$ the subset of the current `candidates` that have exactly best performance on `case`

      **if** *candidates contains only one single* `candidate` **then**

         `parent` $\leftarrow$ `candidate`

         break

      **end**

   **end**

   **if** *parent is None* **then**

      `parent` $\leftarrow$ a randomly selected individual in `candidates`

   **end**

   `candidates` $\leftarrow$ set of $p$ instances of the DNN model copied with the same parameters as `parent`

**end**

**return** *parent*

---

lexicase selection lets all the candidates survive because they all have the same "best" performance, which is failure. However, in the early stages of DNN training, while all the candidates are unable to predict correctly on any case, the original lexicase selection will proceed to evaluate them until someone happens to get a correct prediction by chance, which is inefficient especially on large datasets. So we modify the algorithm to randomly select a candidate from the remaining candidates if they all fail. The modification improves the efficiency of early-stage training and has not shown any noticeable effect on the final model performance.

## 4 EXPERIMENTAL RESULTS

The proposed Gradient Lexicase Selection is tested on the task of image classification, which is one of the most common benchmark problems in computer vision and deep learning in general. We implement the algorithm on six popular DNN architectures (VGG (Simonyan & Zisserman, 2015), ResNet (He et al., 2016), DenseNet (Huang et al., 2017), MobileNetV2 (Sandler et al., 2018), SENet (Hu et al., 2018), EfficientNet (Tan & Le, 2019)). To show the significance of our method, we also implement the original momentum-SGD training as baselines for all the architectures.

Three benchmark datasets (CIFAR-10 (Krizhevsky et al., 2009), CIFAR-100 (Krizhevsky et al., 2009), and SVHN (Netzer et al., 2011)) are used for evaluation. These datasets comprise $32 \times 32$ pixel real-world RGB images of common objects (CIFAR-10, CIFAR-100) and street scene digits (SVHN). The training is done using the training set only and we evaluate the methods on the test set after the training process is finished. Note that for illustration purposes, we only use the training dataset of SVHN without the large *extra* set, so the results are not comparable to other work.

Table 1: Image classification results. We report the mean percentage accuracy (*acc.*) with standard deviation (*std.*) obtained by running the same experiment with three different random seeds. The last column (*acc.* ↑) calculates the difference of accuracy by using our method compared to baseline, where positive numbers indicate improvement.

| Dataset | Architecture | Baseline | | Lexicase | | |
| --- | --- | --- | --- | --- | --- | --- |
| | | *acc.* | *std.* | *acc.* | *std.* | *acc.* ↑ |
| | VGG16 | 92.85 | 0.10 | 93.40 | 0.13 | **0.55** |
| | ResNet18 | 94.82 | 0.10 | 95.35 | 0.06 | **0.53** |
| | ResNet50 | 94.63 | 0.46 | 94.98 | 0.18 | **0.34** |
| CIFAR-10 | DenseNet121 | 95.06 | 0.31 | 95.38 | 0.04 | **0.32** |
| | MobileNetV2 | 94.37 | 0.19 | 93.97 | 0.12 | -0.39 |
| | SENet18 | 94.69 | 0.14 | 95.37 | 0.23 | **0.68** |
| | EfficientNetB0 | 92.60 | 0.18 | 93.00 | 0.22 | **0.40** |
| | VGG16 | 72.09 | 0.52 | 72.53 | 0.20 | **0.44** |
| | ResNet18 | 76.33 | 0.29 | 76.68 | 0.40 | **0.35** |
| | ResNet50 | 76.82 | 0.96 | 77.44 | 0.25 | **0.63** |
| CIFAR-100 | DenseNet121 | 78.72 | 0.82 | 79.08 | 0.26 | **0.36** |
| | MobileNetV2 | 75.87 | 0.28 | 75.57 | 0.30 | -0.30 |
| | SENet18 | 76.97 | 0.06 | 77.22 | 0.29 | **0.25** |
| | EfficientNetB0 | 71.03 | 0.86 | 71.36 | 0.87 | **0.33** |
| | VGG16 | 96.27 | 0.06 | 96.29 | 0.08 | **0.02** |
| | ResNet18 | 96.43 | 0.14 | 96.62 | 0.08 | **0.19** |
| | ResNet50 | 96.69 | 0.21 | 96.74 | 0.07 | **0.04** |
| SVHN | DenseNet121 | 96.82 | 0.16 | 96.87 | 0.03 | **0.05** |
| | MobileNetV2 | 96.23 | 0.13 | 96.26 | 0.07 | **0.03** |
| | SENet18 | 96.62 | 0.19 | 96.59 | 0.11 | -0.03 |
| | EfficientNetB0 | 96.14 | 0.12 | 95.94 | 0.10 | -0.20 |

## 4.1 IMAGE CLASSIFICATION RESULTS

The image classification results are shown in Tab. 1. We report the mean percentage accuracy (*acc.*) with standard deviation (*std.*) obtained by running the same experiment with three different random seeds. The last column (*acc.* ↑) calculates the difference of accuracy by using our method compared to baseline, where positive numbers indicate improvement. We can first see that by using our method, most of the architectures show significant improvement on the testing result. On the easier SVHN dataset, we can still observe moderate and consistent improvement. To show the robustness of our algorithm, we use the same population size of 4 for lexicase in all the experiments, meaning the performance may be further improved if extra tuning is performed. The ablation study on the effect of population size is described later in Sec.5.1.

Beyond those improvements, we also find that among all the architectures, our method surprisingly fails to improve MobileNetV2 on both CIFAR-10 and CIFAR-100. The main difference between MobileNetV2 and other architectures is that it is a highly optimized architecture with over an order of magnitude less parameters compared to other architectures. Sandler et al. (2018) stated that they tailor the architecture to different performance points, which can be adjusted depending on desired accuracy/performance trade-of. As a result, it is likely that the accuracy is restricted by the model size, and even with better training strategies the performance is not going to improve. Such results indicate that our method may not work directly with architectures that have been optimized by using other training methods. But for other more general architectures our method work directly out-of-the-box without further tuning.

## 4.2 COMPARING DIFFERENT SELECTION METHODS

The proposed method has two major components, SubGD and Lexicase Selection. To further validate the contribution of each component, we introduce two other selection methods for compari-

Table 2: Comparing gradient lexicase selection to other selection methods on CIFAR-10. We report the mean percentage accuracy (*acc.*) with standard deviation (*std.*) obtained by running the same experiment with three different random seeds.

| Architecture | SGD | | Random | | Tournament | | Lexicase | |
|---|---|---|---|---|---|---|---|---|
| | *acc.* | *std.* | *acc.* | *std.* | *acc.* | *std.* | *acc.* | *std.* |
| VGG16 | 92.85 | 0.10 | 92.97 | 0.15 | 93.12 | 0.12 | **93.40** | 0.13 |
| ResNet18 | 94.82 | 0.10 | 94.99 | 0.12 | 94.90 | 0.14 | **95.35** | 0.06 |
| ResNet50 | 94.63 | 0.46 | 94.75 | 0.13 | 94.77 | 0.04 | **94.98** | 0.18 |
| DenseNet121 | 95.06 | 0.31 | 95.13 | 0.04 | 95.12 | 0.02 | **95.38** | 0.04 |
| MobileNetV2 | **94.37** | 0.19 | 94.02 | 0.14 | 93.91 | 0.09 | 93.97 | 0.12 |
| SENet18 | 94.69 | 0.14 | 95.04 | 0.15 | 95.01 | 0.23 | **95.37** | 0.23 |
| EfficientNetB0 | 92.60 | 0.18 | 92.77 | 0.11 | 92.83 | 0.12 | **93.00** | 0.22 |

son: random selection and tournament selection (Miller et al., 1995). Using the same evolutionary framework with SubGD, random selection simply selects a random offspring for each generation, and tournament selection uses the average accuracy as the fitness function for selection, which is an aggregated metric as oppose to lexicase selection. The results are shown in Tab. 2.

We can observe that both random selection and tournament selection perform slightly better than the SGD baseline in most cases, but the proposed gradient lexicase selection is consistently better than both methods with a significant margin. Random selection can be viewed as a baseline method that uses SGD with the same amount of computation as gradient lexicase selection, which indicates that the proposed method outperforms SGD even at the same level of computation. Tournament selection is one of the most commonly used selection method in evolutionary algorithms, which select parents based on an aggregated fitness evaluation. As the performance of tournament selection is similar to random selection, indicating that the mechanism of lexicase selection has the major contribution to the improvement.

## 5 ABLATION STUDIES

In this section, we design several ablation studies to further analyze and validate the effectiveness of the proposed method. Unless specifically mentioned, all the implementation details follow the same practices in Sec. B.

### 5.1 POPULATION SIZE

Population size is an essential hyperparameter for evolutionary algorithms. In this work, the population size of DNNs has more constraints such as computation cost and total GPU memory, so it has to be much smaller comparing to those for classic GP problems. We test lexicase gradient selection with population sizes of $2, 4, 6, 8$. For illustration purposes, two architectures (VGG16 and ResNet18) are evaluated on the CIFAR-10 dataset. The results are shown in Tab. 3.

First, we can see that lexicase is relatively robust to different population size $p$. Under all the population size configurations lexicase manages to outperform baseline significantly. Since there have not been a trend of increased accuracy with larger population size, the generalization performance does not seem to increase with a larger population. This observation aligns well with the behavior of lexicase selection in GP problems (Helmuth et al., 2018; La Cava et al., 2019), where there seems to be an optimal population size through the trade-off between exploration and exploitation.

For our method, having a larger population size not only adds more offspring for each generation, but also reduces the size of each subset used for training each offspring by SubGD. In either way, the exploration is reduced and the exploitation is increased. There is no conflict between the two effects, so we do not control the size of each subset when increasing the population size. If the population gets too large, the individuals may not evolve different-enough behaviors from each other, and thus the diversity of population may become lower. In general, we find that with relatively small population size, we can get good results for gradient lexicase selection.

Table 3: Comparing different population sizes on CIFAR-10. Lexicase is relatively robust to different population size $p$, and it manages to outperform baseline with all the configurations in this experiment. The generalization performance does not seem to increase with larger population size.

| Architecture | | Lexicase with population size $p$ | | | |
|---|---|---|---|---|---|
| | Baseline | $p = 2$ | $p = 4$ | $p = 6$ | $p = 8$ |
| VGG16 | 92.85 | 93.61 | 93.40 | **93.92** | 93.37 |
| ResNet18 | 94.82 | **95.50** | 95.35 | 95.27 | 95.38 |

Table 4: Comparing different momentum configurations on CIFAR-10. Resetting momentum after each selection event avoids too much aggregation of gradients, which results in a higher diversity of offspring and thus better generalization performance.

| Architecture | | Lexicase with different momentum options | | |
|---|---|---|---|---|
| | Baseline | No Momentum | Reset Momentum | Inherit Momentum |
| VGG16 | 92.85 | 92.95 | **93.40** | 93.13 |
| ResNet18 | 94.82 | 94.77 | **95.35** | 95.23 |

## 5.2 MOMENTUM

In deep learning, SGD with momentum (Sutskever et al., 2013; Liu et al., 2020) has been one of the most widely adopted methods for training DNNs. Momentum accelerates gradient descent by accumulating a velocity vector in directions of persistent reduction in the objective across iterations. This accumulating behavior actually interferes with the proposed gradient lexicase selection algorithm, because model instances in the population get different gradient updates, and thus will have different momentum parameters.

To solve this issue, we propose three options: 1) No Momentum: do not use momentum at all; 2) Reset Momentum: use high momentum rate and re-initialize the momentum parameters every epoch for each model instance; 3) Inherit Momentum: when selecting the parent model, also copy the momentum parameters along with the model parameters to all the instances in the next generation. For this study, we also test two architectures (VGG16 and ResNet18) with population size of 4 on the CIFAR-10 dataset. The results are shown in Tab. 4.

The key idea of lexicase selection is to select parent by using a sequence of training cases that are prioritized lexicographically for each generation. In this way, the population can maintain a high level of diversity. On the other hand, momentum tends to find an aggregated direction of gradient update accumulated through time. From the results, we can see that the Reset Momentum option works the best, indicating that if we inherit momentum, it will influence the mutation over generations and thus the selection strength of lexicase will be negatively affected. By resetting momentum each epoch, only the mutation in the current generation is accelerated by using momentum SGD, which results in a higher diversity of offspring. In general, the momentum options can also be viewed as different trade-offs of exploration and exploitation.

## 5.3 REPRESENTATION DIVERSITY

While quantitative results have shown that the proposed method manages to improve the generalization of DNNs, we would like to further investigate the reasons behind this in order to better understand the behavior of the algorithm. One hypothesis is that since lexicase selection is able to increase the diversity and generality of the population in GP, it may as well help DNNs learn more diverse representations, which improves the overall model generalization. To validate this, we analyze and compare the feature representations in ResNet-18 trained using normal SGD and gradient lexicase selection. We take the first 100 samples from the CIFAR-10 test set and use global max pooling to obtain the channel-wise activations of `conv_4x` and `conv_5x` layers (as defined in He et al. (2016)). Fig. 2 shows the results.

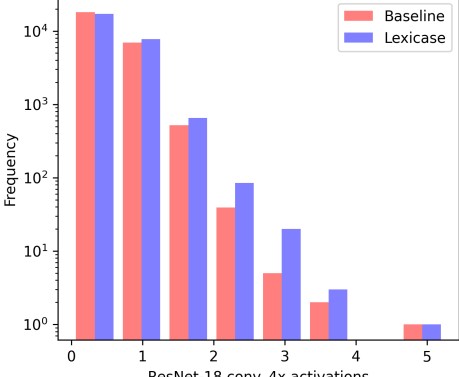 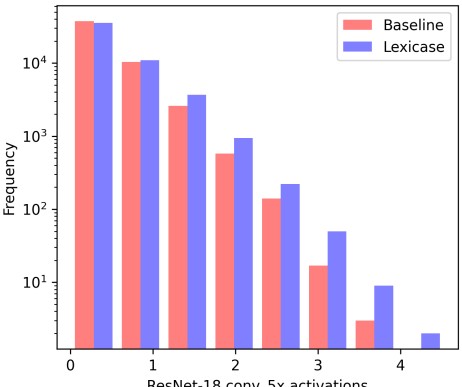

Figure 2: Comparing representation diversity of normal SGD (Baseline in red) and gradient lexicase selection (lexicase in blue). The flatter distribution shows that our method produces more diverse representations.

We can observe that our method produces a flatter distribution of activations with less frequency on 0s and more frequency on other values. Ioffe & Szegedy (2015); Wu & He (2018) shows that normalized distribution of layer activations can help reduce the internal covariate shift of DNNs during training, and thus improves the training efficiency and model generalization. Similarly, our method manages to learn more diverse representations by incorporating lexicase selection into the training framework, which is advantageous to the generalization of DNNs.

## 6 CONCLUSION AND FUTURE WORK

In this work, efficient adaption of lexicase selection in the task of optimizing deep neural networks is explored. We propose gradient lexicase selection, an evolutionary algorithm that incorporates lexicase selection with stochastic gradient descent to help DNNs learn more diverse representations for better generalization. Experimental results show that the proposed method can improve the performance of several popular DNN architectures on benchmark image classification datasets. Several ablation studies further validate the robustness and advantages of our method from different perspectives. More specifically, we investigate the trade-offs between exploration and exploitation by analyzing the effects of population size and momentum. We also show that our algorithm can produce better representation diversity, which is advantageous to the generalization of DNNs.

The goal of our method is to improve the generalization performance rather than speed up the optimization. Our method is potentially valuable to many real-world problems, especially those safety-critical applications like autonomous vehicles, where higher cost of computation during training is acceptable for better generalization performance. There are also several factors to consider regarding the computation cost: 1) our method expects parallel training of model instances, so the optimal training time can be reduced to naive SGD training with modern cloud computing facilities; 2) we have shown that with relatively small population sizes ($4\times$ naive SGD), our method can already achieve significantly better performance; 3) with the same amount of computation, the naive method (random selection baseline in Sec.4.2) can not achieve the same performance as ours.

There are several limitations of our work. As described in Sec. 4.1, the current gradient lexicase selection method may not work with architectures that have been highly optimized, indicating a potential correlation between network architecture and lexicase selection. For future directions, we look forward to explorations on how lexicase selection can be used in optimizing neural architectures along with the parameters, and the integration of lexicase selection in neural architecture search in general.

ACKNOWLEDGMENTS

This material is based upon work supported by the National Science Foundation under Grant No. 1617087. Any opinions, findings, and conclusions or recommendations expressed in this publication are those of the authors and do not necessarily reflect the views of the National Science Foundation.

This work was performed in part using high performance computing equipment obtained under a grant from the Collaborative R&D Fund managed by the Massachusetts Technology Collaborative.

The authors would like to thank Ryan Boldi, Edward Pantridge, Thomas Helmuth, and Anil Saini for their valuable comments and helpful suggestions.

REPRODUCIBILITY STATEMENT

We submit our source code as the supplementary material for the review process, which can be used to reproduce the experimental results in this work. We also release our source code on Github: `https://github.com/ld-ing/Gradient-Lexicase`. Experiment configurations and implementation details are described in Sec. B.

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

## A  LEXICASE SELECTION

Lexicase selection is a parent selection method in population-based stochastic search algorithms such as genetic programming (Helmuth et al., 2014; Spector, 2012). The lexicase selection algorithm is outlined in Alg. 2.

---

**Algorithm 2:** Lexicase selection to select one parent program in genetic programming

---

**Data:**

- `cases` - randomly shuffled sequence of data samples to be used in selection

- `candidates` - the entire population of programs

**Result:**

- an individual program to be used as a parent

**for** *case in* `cases` **do**
  `candidates` ← the subset of the current `candidates` that have exactly best performance on `case`
  **if** *candidates contains only one single* `candidate` **then**
    | **return** `candidate`
  **end**
**end**
**return** *a randomly selected* `candidate` *in* `candidates`

---

The key idea of lexicase selection is that each selection event considers a randomly shuffled sequence of training cases. With the specific ordering, only individuals whose error is minimal among all the already-considered cases are allowed to survive. Moving forward through the sequence of cases, the selection is done when there is only one candidate left or until all the training cases have been gone through, in which case we select randomly from the remaining candidates.

Since the ordering of training cases is randomized for each selection event, every training case gets the opportunity to be prioritized when being put at the beginning of the sequence. As a result, lexicase selection sometimes selects specialist individuals that perform poorly on average but perform better than many individuals on one or more other cases.

## B  IMPLEMENTATION DETAILS

Each network architecture with baseline SGD training and its corresponding counterpart with gradient lexicase selection are trained with identical experimental schemes. We use SGD with momentum instead of the popular adaptive methods (such as Adam) because, despite the popularity of those methods, some recent works (Luo et al., 2018; Wilson et al., 2017) observe that the solutions found by those methods actually generalize worse (often significantly worse) than SGD. We did experiments with Adam and tuned the learning rate for several trials, but the results are significantly worse than the SGD counterpart. This work focuses more on the generalization performance rather than the training speed, so we follow the common practice to use SGD with momentum for both baseline training and SubGD. However, it is very likely that some most recent optimization methods, such as Luo et al. (2018), can achieve faster training as well as the same generalization performance as SGD.

We follow standard practices and perform data augmentation with random cropping with padding and perform random horizontal flipping during the training phase (no augmentation is used during selection phase). The input images are normalized through mean RGB-channel subtraction for all the phases. For both baseline and lexicase, we use SGD with momentum of $0.9$. For lexicase, we use the Reset Momentum option that re-initialize the momentum parameters for each epoch, which is explained in detail later in Sec. 5.2.

The batch size is set to $128$ for CIFAR-10 and $64$ for CIFAR-100 and SVHN. The initial learning rate is set to $0.1$ and tuned by using Cosine Annealing (Loshchilov & Hutter, 2017).

We set the total number of epochs as 200 for baseline training and as $200(p+1)$ for gradient lexicase selection, where $p$ is the size of population. For each epoch in lexicase, the mutation only uses $1/p$ of the training data to train each model instance, so we keep the total iterations of weight update of lexicase training similar to baseline training to ensure convergence.

