# OpenReview forum: "Optimizing Neural Networks with Gradient Lexicase Selection"
_ICLR.cc/2022/Conference — ICLR 2022 Poster_

### Official Review · Reviewer_U4uc · 2021-11-01

**Correctness:** 3
**Technical Novelty And Significance:** 2
**Empirical Novelty And Significance:** 1
**Recommendation:** 6
**Confidence:** 3

**Details Of Ethics Concerns:**

No concern here

**Main Review:**

The training strategy presented in this paper appears expensive in terms of 1) holding multiple copies of the models at all times and 2) looping over permutations of test points for searching the best candidates. That means training this model is as expensive as training multiple independent models. Especially, when the author stated that the proposed method needs to train 200(p+1) epochs, which is slower than training a single model (even with the same number of instance iterations). Compared to the <<1% accuracy improvements, I am curious why people want to use this approach. With the same complexity, one can use the ensemble method to achieve better performance, calibration~(uncertainty estimation), etc.

Experiments also seem to be missing some critical components. For example, since the training strategy optimally searches candidates among the many in the same generation, the entire training process treats data inequivalently. The experiments should show the skewness of data used in the final training model to show the strength of the proposed work. By comparing a model that is directly trained on the weighted data points (weights are from the proposed approach), we can see if the data point's weighting is the reason for the better performance (order independent). This experiment is to identify the real reason for the better model performance. This is just one experiment I can quickly think of, and I think the authors should consider including something like this to enhance their claim.

Some experiment conclusions are not quite convincing—for example, Table 2. I am not sure if the population size P is indeed helpful in this case since the population size difference does not significantly impact the results. What can I learn from this?

The experiments only showed accuracy as the metric. There are two concerns here. 1) Even for accuracy, the proposed method (with the significant sacrifice on training complexity), in some cases, performs poorly compared to the baseline. 2) The proposed method is very greedy. Does it affect the model's calibration? E.g. predicted likelihood mismatch data distribution a lot? What is the overall loss comparison on the validation set?


**Summary Of The Paper:**

This paper presents a neural network training strategy that leverages lexicase selection and evolutionary algorithms. Specifically, the training strategy involves two layers of loops to iteratively update model&select candidates, which searches the best optimization direction through greedy search. Experiments are conducted on two image datasets and show the proposed training strategy allows the learned model to reach better performance (accuracy). Further experiments investigate how hyperparameters would impact the final performance.

**Summary Of The Review:**

1. Please justify where the proposed training strategy is more useful than ensemble models as they have similar complexity.
2. Please include experiments that justify the reason of the performance improvement (see above).
3. Please add comparison between ensemble model and the proposed method in terms of both accuracy and other metrics (Expected Calibration Error, Cross-entropy Loss, etc)
3. Algorithm 1 seems not quite helpful in this paper. Please consider to remove it and make more discussion to justify the claims.

---

> ### Author Response · Authors · 2021-11-20
> **Response to Reviewer U4uc**
>
> Thank you for your valuable feedback and comments! We address the points below to clear up some misunderstandings and convince you of the novelty and contribution of our work.
>
> >training this model is as expensive as training multiple independent models. Compared to the <<1% accuracy improvements, I am curious why people want to use this approach.
>
> You are correct that the computation cost is about training p models in parallel. However, the improvement is indeed significant, given that the baseline accuracies are already high. For example, for VGG 16 on CIFAR-10, our method improves accuracy by 0.55%, which according to the 7.15% error rate is a 7.7% relative improvement. The goal of our method is to improve the generalization performance rather than speed up optimization. This is valuable to many real-world problems, especially safety-critical ones like autonomous vehicles where 0.1% accuracy may save people's lives. There are also several factors to consider beyond the total computation cost: 1) our method expects parallel training of models, so the actual training time can be reduced to naive training in the optimal case; 2) our method can achieve good performance with small population size. To better explain this, we also add more discussion on this in Sec. 6.
>
> >Please justify where the proposed training strategy is more useful than ensemble models as they have similar complexity.
>
> The method proposed is an evolutionary method for optimizing a single model, it is not an ensemble method. The key difference is that our method produces one single model at the end, but ensemble methods use several different models and aggregate their predictions. This will make a huge difference in testing where the inference time of ensemble methods is significantly higher.
>
> >Include experiments to identify the real reason for the better model performance.
>
> The key procedure of lexicase selection can be viewed as a non-numerical weighting of the cases, which can prevent the compromising behavior that may be introduced by aggregated metrics. It is possible that similar performance may be achieved by some kind of dynamic numerical weighting of training cases, and we look forward to extending this direction in future work. For this work, we add the two baseline experiments in the revision, a random selection baseline and a tournament selection baseline (which uses average accuracy as an aggregated metric for fitness evaluation). The results show that both random selection and tournament selection perform slightly better than the baseline but significantly worse than lexicase selection, which further validates the effectiveness of our method. More details are included in Sec. 4.2.
>
> >Table 2. I am not sure if the population size P is indeed helpful in this case since the population size difference does not significantly impact the results.
>
> As explained in Sec. 5.1, the results show that 1) there is a trade-off between exploration and exploitation by altering population size, and the best results are achieved by some middle point. 2) with relatively small population size (less extra computation), we can get good results of lexicase gradient descent and outperforms the baseline method consistently.
>
> >Even for accuracy, the proposed method (with the significant sacrifice on training complexity), in some cases, performs poorly compared to the baseline.
>
> As explained in Sec. 4.1, para 2, we find that among all the architectures, our method surprisingly fails to improve MobileNetV2, which is a highly optimized architecture with over a magnitude fewer parameters. The architecture is tailored to different performance points, so it is very likely that the accuracy is restricted by the model size, and even with better training strategies, the performance is not going to improve. In future work, we aim to further investigate this problem by tuning the architecture along with the weights.
>
> >The proposed method is very greedy. Does it affect the model's calibration?
>
> The proposed method is not essentially greedy. In fact, our method puts more regularization on the model to not optimize only for lower loss, and we can see that the generalization performance is better. In Sec. 5.3, a qualitative analysis of feature representation activations is conducted, and we find that our method produces a flatter distribution of activations with less frequency on 0s and more frequency on other values, which is advantageous to the generalization of DNNs. Since it is not a common practice to compare test loss on image classification, we did not log the loss. Due to the time constraint, we are not able to show the likelihood comparison in the current revision, but we think this is a great suggestion that can help further explore the behavior of our model, and we will add it to the final version.
>
> >Algorithm 1 seems not quite helpful in this paper.
>
> We moved it to Appendix in case people need more background knowledge about lexicase selection.

---

> > ### Comment · Reviewer_U4uc · 2021-11-29
> > **Thanks for the authors response**
> >
> > After reading the authors response to other reviewers, I am convinced the contribution of this paper is not trivial. However, as reading the paper itself, I am still concerned about the computation cost and the value we obtained. I would encourage the authors to include many of the declarations into the manuscript to make it more clear to the readers. I will raise my score.

---

### Official Review · Reviewer_Fzye · 2021-11-02

**Correctness:** 3
**Technical Novelty And Significance:** 3
**Empirical Novelty And Significance:** 2
**Recommendation:** 6
**Confidence:** 5

**Details Of Ethics Concerns:**

No ethics concerns in regards of this submission.

**Main Review:**

The paper is well written and organized. The language is clear at all times, and the proposed method is explained well enough. The literature review is comprehensive and in place.

The only significant drawback in editing style of the draft is the way algorithms are presented. Both algorithms 1 and 2, rely on _while True do_ control flow blocks, along their corresponding _if ... then break_ cases, which makes the algorithms unorthodox in their presentation, and difficult to understand and analyze.

Despite all above, the major flaw of the paper are the empirical assessment and obtained results, that in my opinion, render this work unsuitable for acceptance. I will detail my specific concerns:

- The performance gains observed in the experimental results are simply too marginal. It's been argued that major forums such as ICLR should move away from works that represent marginal improvements over baseline or state of the art methods, and instead favor innovative approaches, or methods that significantly outclass existing research.

- The results might be even deceptive. Authors never explicitly state that baseline SGD was given the same computational resources (CPU threads-GPU cores-exec. time) than that of their method. They hint at the idea by mentioning that some actions were taken in order to guarantee "same number of evaluations", but that is not enough to consider it a fair comparison. The computational resources should be the same (CPUorGPU-time)- because if that is not the case, and with such marginal improvements, it could be the case that by simply running SGD a few more epochs, their proposed approach could be left behind.

- One part of their proposed approach, the lexicase evaluation, worries me because it seems to be a choke point for the algorithm running time, that could leave their method in serious disadvantage against SGD. Hardware has moved towards SIMD architectures, which has allowed us to harness the power of algorithms such as SGD, and even population-based method to certain extent; however, their approach would require a MISD architecture to be efficiently implemented, which is, to my knowledge, nonexistent, (or at least not mainstream).

Although the proposed approach seems interesting enough, it is simply no attractive enough to switch from SGD. My suggestion for the authors is to search for cases where their method can excel and surpass most other baselines. Exactly as they hinted at the beginning of their paper, perhaps scenarios where data is scarce or incomplete, could be a better test ground for their proposed method. Or simply go directly for NAS and other problems where SGD is known to fail, (e.g. heterogeneous networks).

++UPDATED SCORE 29TH NOV++
After authors replies and improved version of the paper, I raise my score. I think the proposed approach is very interesting; the empirical assessment looks more like something of a work in progress, but authors definitively seem to be on to something.

**Summary Of The Paper:**

Authors propose a method to optimize deep networks through a combination of gradient descent and a population-based mechanism. In particular, authors propose an adaptation of "lexicase" selection, an exotic selection mechanism proposed in evolutionary computation, in order to be used in the context of deep learning and deep networks optimization. In gross terms, _lexicase selection_ consists of picking individuals for surviving and/or as parents for generating offspring, based on their performance test-case instance-wise, rather than based on some aggregated metric such as MSE or loss, as usually done. Authors provide results from a extensive battery of tests where they compare the performance of their method, against that of regular SGD, optimizing several well-know deep architectures in a set of commonly used too benchmark datasets.

**Summary Of The Review:**

The performance improvements over baseline methods are too marginal, nor the method is novel enough, for the work to be considered above acceptance threshold.

---

> ### Author Response · Authors · 2021-11-20
> **Response to Reviewer Fzye**
>
> Thank you for your valuable feedback and criticism. In the following, we address each point individually. While it seems that we disagree on several key issues, we hope our answer can clear up some misunderstandings and clarify the novelty and contribution of this work.
>
> >The only significant drawback in the editing style of the draft is the way algorithms are presented.
>
> We modified both algorithms to use For loops instead of While loops as suggested.
>
> >The performance gains observed in the experimental results are simply too marginal.
>
> The improvement of our method is indeed significant, given that the baseline accuracies are already high. For example, for VGG 16 on CIFAR-10, our method improves accuracy by 0.55%, which according to the 7.15% error rate is a 7.7% relative improvement. Furthermore, the experiments show that the improvement is consistent across different architectures and different datasets (without any specific tuning), demonstrating the generality and robustness of our method.
>
> >Authors never explicitly state that baseline SGD was given the same computational resources (CPU threads-GPU cores-exec. time) than that of their method.
>
> The baseline SGD method trains the network to convergence (got 100% training accuracy at ~150 epoch of 200), so running SGD a few more epochs is unlikely to improve the accuracy. However, we do see that this concern is reasonable and may need further validation. In the revision, we add the two baseline experiments, a random selection baseline, and a tournament selection baseline. The random selection baseline uses the exact same amount of computation as our method, but just replaces the lexicase selection part with random selection, which can be viewed as "training SGD for the same amount of epochs". The tournament selection is running selection over average accuracy, which refers to "running the same amount of epochs for training and selecting the best model according to the training accuracy". The results show that both random selection and tournament selection perform slightly better than the baseline but significantly worse than lexicase selection. We hope these comparisons help convince you of the effectiveness of our method. More details are included in Sec. 4.2.
>
> >however, their approach would require a MISD architecture to be efficiently implemented, which is, to my knowledge, nonexistent, (or at least not mainstream).
>
> The actual computation cost regarding hardware and architecture is definitely an important factor in modern machine/deep learning. While the selection part of our method acts in an MISD fashion, it can be parallelized across different GPUs and each GPU can get its own copy of the data stream for faster inference. Moreover, the main bottleneck of our method is actually the training part with subset gradient descent, which can also be paralleled with multi-GPU training, so the actual training time can be reduced to the same scale of naive training in the optimal case. In this work, our goal is to improve the generalization performance rather than speed up the optimization, and we think this is valuable to many real-world problems, especially safety-critical applications like autonomous vehicles. In addition, experiments in Sec. 5.1 shows that with a relatively small population size, our method can already achieve significantly better performance than naive training, and Sec. 4.2 shows that with the same amount of computation, the naive method can not achieve the same performance as ours. To better explain the actual cost of computation, we add more discussion in Sec. 6 and Sec. 7.2.

---

> > ### Comment · Reviewer_Fzye · 2021-11-26
> > **Thanks for authors response**
> >
> > I'd like to thank authors for their detailed response addressing my concerns raised during my initial review of their work.
> > I also would like to acknowledge their improved version of their draft.
> >
> > After reading their response, as well as the responses to other reviewers, I think I now understand better the motivation behind their proposed method.
> >
> > I am considering upgrading my score, but I still have some doubts regarding the experimental study presented in their paper. If you could please elaborate a bit on the following issue, it would certainly help me reach a final decision:
> >
> > - Why did you design the experimental study in such way? What was your logic behind such wide set of tests across many different architectures and datasets? I ask this because it looks to me somewhat "general purpose" kind of empirical assessment. As you have put forward, the true motivation behind your method is to achieve better generalization, in which case, I'd have expected different experiments. When trying to find evidence of a method that generalizes better, I can imagine three standard types of tests:
> >
> > 1. Show training accuracy vs test accuracy, to remark that the proposed method tends less towards overfitting and/or memorization. Which is kind of done, since you mention standard SGD achieves 100% training accuracy, but it is kind of implicit, and not remarked through the paper.
> > 2. An ablation study decreasing training dataset size, thus showing method's higher data efficiency. This could also be done with datasets known to require data augmentation.
> > 3. Contaminating datasets with increasing level of noise is another standard experiment for methods aimed at obtaining better generalization.
> >
> > Performing such kind of experiments with a couple of different architectures and the same training datasets you selected, would have made more sense to me, if you were aiming for better generalization, rather than test a wide variety of architectures but only superficially. So I am really interested in hearing your line of though for such matter.

---

> > > ### Author Response · Authors · 2021-11-27
> > > **Follow-up Response to Reviewer Fzye**
> > >
> > > Thank you for your response! We are glad to hear that the previous response and revision have been helpful, and we address your follow-up questions as follows:
> > >
> > > >What was your logic behind such a wide set of tests across many different architectures and datasets?
> > >
> > > In this work, our goal is to establish a proof-of-concept that the idea of lexicase selection can also improve the generalization (i.e., testing performance) of deep neural networks, efficiently using the proposed evolution framework. Since this idea is new to the field, we choose to test it on the very common image classification benchmarks with different popular network architectures. There are two reasons:
> > > 1. Generality: While there are a few popular network architectures (like the ones used in this work), in practice, people are likely to use custom architectures based on factors like computation resource, size of training data, inference speed requirement, etc. So showing our method works only for one or two architectures/datasets may not be enough to convince people that the proposed method is indeed useful, which will make this work less convincing and less helpful to the community. For the different architectures we use in this work, we intentionally choose architectures with different mechanisms (residual learning, SE activation, etc.) and different sizes (lightweight, normal, and deep networks), and do not use similar variants (such as ResNext), in order to show that our method works for most kind of modern deep networks regardless of their sizes and underlying methodologies. We will add more description to our general experiment design idea in the final version.
> > > 2. Potential Benefits: Since it has been shown that network architectures can be easily adapted to different tasks (e.g., Mask-RCNN shows that using a better image classification network as the backbone improves the performance of object detection), we do want to test the proposed method for "general purpose", in which way to show not only the effectiveness of our method but also the potential benefit to various other related ML tasks. Overall, we hope this work establishes a good foundation and we definitely would like to explore those topics in future work. We will add more discussion on the future directions as well.
> > >
> > > >Show training accuracy vs test accuracy, to remark that the proposed method tends less towards overfitting and/or memorization.
> > >
> > > In this work, the training accuracy of all models on CIFAR-10 gets 100% or 99.9%. Since most modern network architectures are able to get 100% training accuracy [1], we assume the test accuracy itself shows the generalization performance. We will elaborate more on this in the experiment part.
> > >
> > > >An ablation study decreases training dataset size, thus showing the method's higher data efficiency.
> > >
> > > Thanks for the suggestion. Systematic exploration on the topic of data efficiency and handling missing data is exactly one of our future directions. For this work, we will add a pilot study on using 25% and 50% of the training data of CIFAR-10 in the final version to further analyze the proposed method on data efficiency.
> > >
> > > >Contaminating datasets with increasing levels of noise is another standard experiment for methods aimed at obtaining better generalization.
> > >
> > > This is also a great suggestion. First, we do want to point out that having better generalization performance on noisy data (which usually refers to weakly-supervised learning) is a different kind of "generalization" as considered in this work. We would like to clarify that the term "generalization" in this work refers to the testing performance, more specifically, to minimize the "generalization error", i.e., difference between “training error” and “test error” [1], rather than the "robustness" to handle different training situations. We will make this point more clear in the introduction.
> > >
> > > Regarding this experiment, we fully agree with you that the ability to handle noisy data is important, especially for real-world applications where training data may have lower quality. While our method is not intentionally designed for handling weakly-supervised training situations, an evaluation on this can further validate the robustness of our method. In the final version, we will test our method on an additional experiment with noisy labels (e.g., 5%, 10%, 20% of the training labels are randomly chosen) and see how that may affect the performance.
> > >
> > > [1] Understanding Deep Learning Requires Rethinking Generalization, Zhang, et al, ICLR 2017

---

### Official Review · Reviewer_ezPJ · 2021-11-04

**Correctness:** 3
**Technical Novelty And Significance:** 3
**Empirical Novelty And Significance:** 2
**Recommendation:** 6
**Confidence:** 4

**Main Review:**

[Strengths]
- This paper is well-written. The proposed algorithm is easy to follow.
- The experimental results show that the proposed gradient lexicase selection improves the CNN performance.

[Weaknesses]
- The reviewer suspects whether the performance improvement is really caused by the gradient lexicase selection. The proposed method modifies the overall procedure of network training to inject the concept of "population" into deep neural network training. That is, several networks are trained in parallel by using different portions of datasets in one epoch. After the lexicase selection, $p-1$ candidate networks are discarded. Therefore, we cannot exclude that the reason for the performance improvement by the proposed method is caused by such parallel training. In summary, the reviewer thinks that the following baseline algorithms should be considered to highlight the effectiveness of the gradient lexicase selection.
    1. The algorithm using random selection for the parent selection in Algorithm 1 instead of the gradient lexicase selection
    1. The algorithm using aggregated loss for the parent selection in Algorithm 1 instead of the gradient lexicase selection
- As the proposed method maintains a population of deep neural networks, the memory consumption is higher than single training.
- It would be better if tasks other than image classification were examined to show the generalization of the proposed method.

[Comments]
- The actual computational times of the proposed method and naive CNN training should be reported. It seems that the computational cost of the proposed method is about $p$ times higher than that of naive training because the authors set the number of training epochs to $200(p + 1)$. The authors should consider the comparison under the same computational cost for a fair comparison.
- The proposed method maintains several deep neural networks in training. The population-based training (e.g., the following literature) may be related to this paper.
    - Xiaodong Cui, Wei Zhang, Zoltán Tüske, and Michael Picheny, "Evolutionary stochastic gradient descent for optimization of deep neural networks," NeurIPS 2018
    - Max Jaderberg, et al., "Population Based Training of Neural Networks," arXiv:1711.09846

**Summary Of The Paper:**

This paper extends the lexicase selection developed in the genetic programming community to apply the idea of lexicase selection to deep neural network training. In the proposed gradient lexicase selection, a parent network is selected based on the lexicase selection from several candidate networks trained using different subsets of the dataset. The authors apply the proposed method to several well-known convolutional neural networks (CNN) and show that the gradient lexicase selection can improve the performance of trained models.

**Summary Of The Review:**

This paper is easy to follow. However, the current experimental evaluation is insufficient to clearly show the effectiveness of the gradient lexicase selection.

---

> ### Author Response · Authors · 2021-11-20
> **Response to Reviewer ezPJ**
>
> Thank you for your valuable feedback and constructive comments! We hope that our answers below can clarify some concerns and convince you of the novelty and contribution of this work.
>
> > Therefore, we cannot exclude that the reason for the performance improvement by the proposed method is caused by such parallel training.
>
> We appreciate your advice and agree that more baseline experiments can help better validate the effectiveness of our method. In the revision, we add the two baseline experiments as suggested, a random selection baseline and a tournament selection baseline (which uses average accuracy as an aggregated metric for fitness evaluation). The results show that both random selection and tournament selection perform slightly better than the baseline but significantly worse than lexicase selection, which further validates the effectiveness of our method. More details are included in the added Sec. 4.2. In general, simply training many model instances in parallel does not improve the model performance, unless further ensembling is introduced. Our method produces one single model at the end, so the improvement does not come from the parallel training.
>
> >As the proposed method maintains a population of deep neural networks, the memory consumption is higher than single training.
>
> This is correct. Our method expects parallel training of a population of model instances, so although total memory consumption is higher, we do not require more memory on every single GPU. The training procedure is similar to training the same model on multiple GPUs, but a selection step is used to replace the normal gradient aggregating step.
> It would be better if tasks other than image classification were examined to show the generalization of the proposed method.
> As a proof of concept, we choose to work on a general and widely studied deep learning problem, image classification, for which the solutions have been shown to generalize to many other tasks as the backbone. We look forward to extending to other more specific tasks in future works.
>
> >The authors should consider the comparison under the same computational cost for a fair comparison.
>
> You are exactly correct that the computation cost is about p times higher than naive training. The goal of our method is to improve the generalization performance rather than speed up the optimization, and we think this is valuable to many real-world problems, especially safety-critical applications like autonomous vehicles. There are also several factors to consider regarding computation beyond the total cost itself: 1) our method expects parallel training of models, so with modern cloud computing environments, the actual training time can be reduced to naive training in the optimal case; 2) we have shown that with relatively small population size, our method can already achieve significantly better performance than naive training; 3) with the additional experiments in Sec. 4.2, we show that with the same amount of computation, the naive method can not achieve the same performance as ours. To better explain this point, we add more discussion on this in Sec. 6.
>
> >The population-based training (e.g., the following literature) may be related to this paper.
>
> We add more references to Sec. 2. to cover the broader topic of using population-based training for purposes such as speeding up evolution (Cui, et al.), hyperparameter tuning (Jaderberg, et al.), etc.

---

> > ### Comment · Reviewer_ezPJ · 2021-11-27
> > **Comments after the author responses**
> >
> > Thank you for your responses. I have checked the additional experimental results. The experimental result regarding the comparison against random selection and tournament selection (Section 4.2) convinces me and shows the effectiveness of the proposed Lexicase selection. The only concern is that the performance gain is small compared to the increase in the computational cost. However, the performance improvement caused by the proposed Lexicase selection became clear after the revision. Therefore, I raise my score.

---

> > > ### Author Response · Authors · 2021-11-27
> > > **Follow-up Response to Reviewer ezPJ**
> > >
> > > Thanks for your comments and previous suggestions on additional experiments! We are glad to learn that the additional experiments have been helpful, and want to make a quick follow-up clarification on your concern.
> > >
> > > > The only concern is that the performance gain is small compared to the increase in the computational cost.
> > >
> > > The improvement of our method is indeed significant, given that the baseline accuracies are already high. For example, for VGG 16 on CIFAR-10, our method improves accuracy by 0.55%, which according to the 7.15% error rate is a 7.7% relative improvement. Furthermore, the experiments show that the improvement is consistent across different architectures and different datasets (without any specific tuning), demonstrating the generality and robustness of our method.
> > >
> > > Also, as mentioned in the previous reply, the goal of our method is to improve the generalization performance rather than speed up the optimization, and we think this is valuable to many real-world problems, especially safety-critical applications. We definitely look forward to reducing the computation cost in future work.

---

### Official Review · Reviewer_bGuc · 2021-11-08

**Correctness:** 4
**Technical Novelty And Significance:** 3
**Empirical Novelty And Significance:** 4
**Recommendation:** 8
**Confidence:** 3

**Main Review:**

The paper is very well-written, flows smoothly and is a pleasure to read. The ideas are well articulated and clear.

The lexicase selection method proposed in the paper seems analogous to ES methods where an area around the current solution is explored and then a selection step leverages the information received from this exploration step to improve the existing solution. The distinctive part here is the use of bagged sub-gradients instead of random mutation operators and distributed selection instead of a centralized one with an aggregate metric. Including ES-based baselines (such as CMA-ES or a more performant variant) would strengthen the claims of the paper.

The baseline uses a standard sgd method with momentum. How does the performance compare against a more modern optimizer? For instance, Adam is often preferred over sgd + momentum as it is simple and tends to find the global optima with good regularity. A comparison integrating Adam would answer questions on how fundamental the merits of the proposed lexicase selection method are? And could it translate across varying choices of optimizers to help practitioners.

On the relationship between population size and performance, should this design choice be informed by the amount of training data available? Since the population bags the data to train, a larger population can mean less data per sub-gradient step. I presume this could affect the exploration/exploitation tradeoff quite noticeably. It would be great if the authors could discuss this in more detail and suggest/provide more insight into how this tradeoff could be balanced without relying on parameter tuning.


**Summary Of The Paper:**

The paper applies lexicase selection towards training deep neural networks using a hybrid evolutionary-gradient descent optimization method. The mutation operator is replaced by sub-gradient descent on bagged data and lexicase selection is used as the selection procedure in the evolutionary framework. Results in a variety of benchmark image classification tasks demonstrate that the proposed method can improve upon standard stochastic gradient descent methods.

**Summary Of The Review:**

The paper presents an interesting evolutionary framework, where mutation is replaced by bagged sub-gradient descent and lexicase selection is used to refresh the population. The proposed method builds on ideas of lexicase selection from related domains and applies it successfully to train deep neural networks on benchmark image classification tasks demonstrating improved performance. The work is novel and potentially valuable to the community.

---

> ### Author Response · Authors · 2021-11-20
> **Response to Reviewer bGuc**
>
> Thank you for your positive review and valuable feedback! The points you made are exactly relevant to our work, and we address them individually as follows.
>
> > Including ES-based baselines (such as CMA-ES or a more performant variant) would strengthen the claims of the paper.
>
> In this work, we aim to establish a proof of concept that the lexicase selection method, which has been used in GP for solving uncompromising problems, can also benefit modern machine/deep learning where an aggregated metric may cause the compromising behavior that undermines the generalization performance. Our next goal is exactly to evaluate other black-box optimization methods (such as CMA-ES) with the proposed evolutionary training framework. To strengthen the claims of this work, we add the tournament selection baseline, which uses an aggregated metric as opposed to the lexicase selection, to better validate the contributions of both the evolutionary framework with SubGD and lexicase selection. The results show that tournament selection performs slightly better than the baseline but much worse than lexicase selection. More details are included in Sec. 4.2 in the revision.
>
> > How does the performance compare against a more modern optimizer? For instance, Adam is often preferred over sgd + momentum as it is simple and tends to find the global optima with good regularity.
>
> Despite the popularity of adaptive methods (including Adam), some recent works [1,2] observe that the solutions found by those methods actually generalize worse (often significantly worse) than SGD. We did experiments with Adam and tuned the learning rate for several trials, but the results are significantly worse than the SGD counterpart. Indeed, many state-of-the-art works (such as [3]) still choose to use SGD with momentum. This work focuses more on the generalization performance rather than the training speed, so we follow the common practice to use SGD for both baseline training and subset SGD mutation. However, it is very likely that some most recent optimization methods, such as [2], can achieve faster training as well as the same generalization performance as SGD. In the revision, we added a paragraph in Sec. 7.2 to illustrate this point.
>
> > On the relationship between population size and performance, should this design choice be informed by the amount of training data available? Since the population bags the data to train, a larger population can mean less data per sub-gradient step.
>
> You are exactly right. Having a larger population size in the proposed framework not only adds more offspring for each generation, but also reduces the size of each subset used for training each offspring. In either way, the exploration is reduced and the exploitation is increased. There is no conflict between the two effects, so we do not control the size of each subset when increasing the population size. In other words, if the size of subsets is controlled, there will be overlaps between subsets when use to train the offspring, which will likely decrease the diversity of the population and introduce redundancy in mutation. So overall this tradeoff can be balanced by having the right population size, and the experiments in Sec. 5.1 shows that different architecture has different optimal population sizes (and it's not larger the better). In the revision, we added more description on this point in Sec. 5.1.
>
> [1] The Marginal Value of Adaptive Gradient Methods in Machine Learning, Wilson, et al., NeurIPS 2017.
>
> [2] Adaptive Gradient Methods with Dynamic Bound of Learning Rate, Luo, et al., ICLR 2019.
>
> [3] High-Performance Large-Scale Image Recognition Without Normalization, Brock, et al., arXiv 2021

---

### Decision · Program_Chairs · 2022-01-20

**Decision:**

Accept (Poster)

**Comment:**

The authors propose bringing lexicase selection from evolutionary computation and applying it to the optimisation of gradient descent. This is done by training a set of p networks and using their performance to select this set of p networks as training progresses on random subsets of the training data.
The reviewers felt, and I agree, that the paper was well written and its method now well described. Concerns raised during review include: additional computational cost, novelty, and marginal performance improvements. Nonetheless after discussion, while the computational cost is indeed higher, it is a novel application, and the reviewers were all in agreement with acceptance after further discussion around experiments.